# *p*-Coumaric Acid Nanoparticles Ameliorate Diabetic Nephropathy via Regulating mRNA Expression of KIM-1 and GLUT-2 in Streptozotocin-Induced Diabetic Rats

**DOI:** 10.3390/metabo12121166

**Published:** 2022-11-23

**Authors:** Amalan Venkatesan, Anitha Roy, Srinivasan Kulandaivel, Vijayakumar Natesan, Sung-Jin Kim

**Affiliations:** 1Department of Biochemistry and Biotechnology, Faculty of Science, Annamalai University, Annamalainagar 608002, Tamil Nadu, India; 2Department of Pharmacology, Saveetha Dental College and Hospitals, Saveetha Institute of Medical and Technical Sciences, Chennai 600077, Tamil Nadu, India; 3Department of Pharmacology, Saveetha College of Allied Health Sciences, Saveetha Institute of Medical and Technical Sciences, Thandalam, Chennai 602105, Tamil Nadu, India; 4Department of Pharmaceutical Chemistry, Nandha College of Pharmacy, Erode 638052, Tamil Nadu, India; 5Department of Pharmacology and Toxicology, Metabolic Diseases Research Laboratory, School of Dentistry, Kyung Hee University, Seoul 02447, Republic of Korea

**Keywords:** diabetic nephropathy, urinary NAG, β-glucuronidase, KIM-1, glucose transporter-2 (GLUT-2)

## Abstract

Diabetic nephropathy (DN) has become a leading cause of end-stage renal failure worldwide. The goal of the current study was to examine the protective effects of chitosan-loaded *p*-Coumaric acid nanoparticles (PCNPs) in nephrotoxicity induced by streptozotocin (STZ). Because of the antidiabetic, anti-inflammatory, and antioxidant properties of PCNPs, the development of DN may be considerably decreased. In this study, the rats received a single intraperitoneal injection (i.p.) of STZ (45 mg/kg) to induce DN. PCNPs were given orally 80 mg/kg b.w to the rats for a duration of four weeks. Body weight, kidney weight, blood glucose, and insulin levels were measured at the end of the experiment. Serum and urine parameters were also examined, along with the histological, immunobiological, and tumor necrosis factor (TNF) and interleukin-6 (IL-6) expression of the nephrotic rats. To comprehend the impact of PCNPs, the expression patterns of the kidney injury molecule (KIM-1) and glucose transporter-2 (GLUT-2) were evaluated. Administration of PCNPs significantly increased body weight, decreased kidney weight and also ameliorated blood glucose levels in the nephropathic rats. The administration of PCNPs also reverted the levels of urea, serum creatinine, urinary NAG, β-glucuronidase and albumin to near-normal levels. The administration of PCNPs also caused the levels of serum and urine parameters to return to near-normal levels. Additionally, the PCNP-treated rats had markedly reduced TNF-α, IL-6, and KIM-1 expressions as well as enhanced GLUT-2 mRNA expression. Our findings clearly showed that PCNP administration prevents the onset of DN in rats by lowering hyperglycemia, decreasing inflammation, and improving the expression of GLUT-2 mRNA in nephropathic rats.

## 1. Introduction

Diabetes mellitus (DM) is a severe and ubiquitous chronic illness globally. It results in both microvascular and macrovascular consequences, including peripheral vascular disorders, neuropathy, retinopathy, nephropathy, stroke and heart disease. It can also promote hypertrophy and basement thickness of the glomerular membrane [1]. DN affects around one-third of diabetic people. By 2050, there will likely be significant growth in the number of diabetic people, which will increase the incidence of diabetic kidney disease or end-stage renal disease (ESRD) [2]. The development of nephropathy is mostly caused by inadequate glycemic management and the accumulation of advanced glycation end products (AGEs). Glomerular function gradually deteriorates in diabetic nephropathy patients [3]. The most successful therapies for advanced DN to date, according to reports, are antihypertensive drugs; however, these are unable to stop the development of DN [4,5,6].

Urinary N-acetyl-/3-D-glucosaminidase (NAG) and β-glucuronidase are higher in DN patients [7]. Diabetes patients may see a rise in urine NAG, which is a general indicator of blood glucose management. In contrast to recently diagnosed diabetics, who both had considerably greater urinary NAG levels than nondiabetic controls, treated diabetics had decreased urinary NAG levels [8]. The initial inflammatory processes in STZ-mediated cellular damage are thought to be due to the generation of TNF-α, inflammatory interleukins, KIM-1, and COX-2.

According to the most recent survey, there are numerous medications available to treat DN, but they all have undesirable side effects and are highly expensive. Therefore, it is essential to investigate DN treatment options based on the various gene expression levels associated with nephropathy [9]. For the development of potent therapeutic agents, plant-derived compounds play a significant role [10]. Various human ailments have been treated with herbal medications. Additionally, the demand for herbal medications such as Ayurvedic, Siddha, and Unani, among others, that are used to treat the disease is rising daily [11,12].

Because herbal-based nutraceutical products have many pharmacological benefits and low toxicity, recent investigations have concentrated on their nephroprotective qualities [13]. *p*-Coumaric acid (*p*-CA) is one of the isomers of hydroxycinnamic acid, which is phenolic nature and is derived from many fruits and vegetables such as potatoes, beans, apples, pears, and beverages such as beer, tea, and chocolate [14,15,16]. p-CA has numerous pharmacological activities such as antitumor, strong anti-inflammatory, antioxidant, anti-angiogenic, antimicrobial, antidiabetic and immunomodulatory activities [11]. Various research has shown that the phenolic component significantly reduced inflammatory processes and cell proliferation by strengthening antioxidant and immune system levels in a variety of experimental scenarios. The ineffective systemic transport and low absorption of these phenolics and phytochemicals in in vivo circumstances, however, is a significant disadvantage [17]. As a result, attempts to apply disease to humans have had mixed results.

According to a recent study, polymeric nanoparticles are superior sources for the synthesis of chemotherapeutic drugs that can be used to treat a broad range of disorders [18]. The nanoparticles based on polymers have largely inclined the controlled and targeted drug-delivery, as well as the biodegradable and biocompatible concepts. Chitosan is one of many biodegradable polymers, and because it is highly biodegradable, biocompatible, and less cytotoxic than other materials, it has been widely utilised to encapsulate therapeutic medicines. Additionally, the nanoparticle drug delivery strategy has been successfully used for a number of drugs, which allows the compound’s stability, slow release, and targetability to be significantly adjusted. Our previous study reported on the synthesis of PCNPs, and their efficiency in in vitro and in vivo antidiabetic activities were studied [11,19] as alternative medicine. An innovative approach called nanomedicine has been proposed to increase the drug’s bioavailability and controlled release.

This study’s hypothesis is that PCNPs can effectively treat renal disorders and chemically induced toxicity. As a result, the current study’s goal was to assess any potential for PCNPs to protect against nephrotoxicity in a male albino Wistar rat model.

## 2. Materials and Methods

### 2.1. Chemicals

Streptozotocin, *p*-Coumaric acid, chitosan, and polyvinyl alcohol (PVA) were purchased from India’s Sigma-Aldrich Chemical Pvt. Ltd., Bangalore, India. The analytical-grade chemicals that were utilized for all other purposes were acquired from Hi-media Mumbai, India.

### 2.2. Preparation of p-Coumaric Acid-Loaded Nanoparticles

PCNPs were prepared by the nanoprecipitation method in the ratio of 1:10:10 (p-CA: Chitosan: polyvinyl alcohol). In 25 mL of ethanol, we dissolved 500 mg of chitosan and 50 mg of *p*-CA. This solution was promptly injected into a solution containing 75 mL of ethanol with 500 mg of polyvinyl alcohol. Then, the solutions were homogenized for 40 min at 18,000 rpm. The remaining fraction was lyophilized to eliminate ethanol and then placed in a rotary vacuum evaporator at 40 °C [11].

### 2.3. Animals

Adult male albino Wistar rats (120–150 g) were received from the central animal house at Annamalai University, Tamil Nadu, India. In addition to having an unlimited daily diet of pellets and access to water, experimental animals were scrupulously cared for in sterile settings at 23 ± 2 °C, 40.5% relative humidity, 12-h light/dark cycle, and alternate. All work was carried out under the guidelines of CPCSEA (AU-IAEC/PR/1306/12/21; dated: 27 December 2021).

### 2.4. Induction of Diabetes Nephropathy

The rats received a single intraperitoneal (i.p.) injection of freshly prepared STZ in 0.1 M citrate buffer after being fasted overnight (pH 4.5). One week after the administration of STZ i.p., glucose levels were monitored using a glucose meter (ACCU-CHEK advantage). Rats were adopted for future investigations by starting the treatments at four weeks when their blood glucose level was above 300 mg/dL [20].

### 2.5. Experimental Design

After a week of adaptation, rats were divided into four groups (n = 6). The experimental rats received PCNPs orally through an intragastric tube once a day for four weeks. The first group consisted of normal rats that were administered vehicle alone. The second group consisted of rats administered STZ (45 mg/kg b.w) in a single-dose i.p. injection. The third group consisted of rats that received STZ (45 mg/kg b.w) in a single-dose i.p. injection and PCNPs (80 mg/kg b.w/day) orally for 28 days. The fourth group consisted of rats that received only PCNPs (80 mg/kg b.w/day) for 28 days. Blood, liver, pancreas and kidney samples were collected for the biochemical and molecular studies after the rats were sacrificed. All of the rats were placed individually in metabolic cages at the end of the study, and 24 h urine samples were taken [20]. To get rid of all the debris, the obtained samples were centrifuged at 1200× *g* at room temperature.

After 12 h of fasting, the rats were sedated by ketamine injection (65 mg/kg/i.p.), and their blood samples (3 mL) were taken by the cardiac puncture into either plain or EDTA-containing tubes and then centrifuged (1200× *g* for 10 min at room temperature) [21]. For further biochemical examination, the retrieved urine, plasma, and serum were kept at −20 °C. The organs (kidney, liver, and pancreas) were then excised from each rat and preserved in ice-cold formalin for histological and immunohistological examination.

### 2.6. Biochemical Assays

A dry, clean test tube was used to collect the blood sample, which was then left to clot for 30 min at room temperature. Centrifugation was used to detach the serum for 10 min at 2000 rpm. The values of glucose, insulin, urea, and creatinine in normal and experimental animals were analyzed by the diagnostic kit; measurements were made for urea using the Natelson et al. method [22], and creatinine was measured using the alkaline picrate method [23]. Using Lockwood and Bosmann’s spectrophotometric technique, urine NAG and β-glucuronidase levels were assessed [19].

### 2.7. Histopathological Examination

The pancreas, kidney, and liver samples were fixed in 10% formalin buffer. The samples were prepared and then embedded in wax after fixing. Using a Leica microtome, tissue slices were stained with hematoxylin and eosin. After the process, the slides acquired this way were examined histopathologically.

### 2.8. Immunohistochemical Examination

The sections of kidney tissues were rehydrated by the grade of ethanol to distilled water. The inclusion of H_2_O_2_ (3%) for 15 min eliminated the activity of endogenous peroxidase. At 37 °C, tissue slices were treated with a blocking solution for 15 min. Sections were correctly labeled with the secondary antibody using horseradish peroxidase after being treated with the appropriate primary antibody against TNF-α and IL-6. Hematoxylin staining was used to see the slides once the color intensity was attained.

### 2.9. RT-PCR

The CFX96 Touch Real-Time PCR Detection System—Bio-Rad CA was used for real-time PCR [24]. To prepare the reaction mix (10 μL), 5 μL of 2× reaction buffer, 0.1 μL of sense and anti-sense primer, 1 μL of cDNA, and 3.8 μL of sterile water were added. Initial denaturation was carried out at 95 °C for 3 min, followed by 40 cycles of PCR, denaturation at 95 °C for 10 s, annealing at 60 °C for 20 s, and extension at 72 °C for 20 s. With no template control (NTC), all reactions were performed in triplicate. The analysis was performed with the Bio-Rad CFX96 Touch Real-Time PCR Detection System and β-actin as a control (Table 1).

### 2.10. Statistical Analysis

The results from the observation are expressed as mean ± standard deviation (SD) of the six rats in each group. One-way analysis of variance (ANOVA) and Duncan’s multiple range test were used to assess the data. The Statistical Package for Social Science (SPSS), Version 23.00, was used to conduct statistical analysis; the significance was defined as *p* < 0.05.

## 3. Results

### 3.1. Body Weight

The body weight of rats was measured in the fourth week of the experiment. Nephropathic rats dropped a lot of weight when compared to healthy rats. The PCNPs (80 mg/kg) treatment was significantly higher compared to DN (Table 2). 

### 3.2. Kidney Weight and Hypertrophy

Significantly greater kidney weight and kidney hypertrophy index were assessed in STZ-induced nephropathic rats than in the normal group of rats, as shown in (Table 2), whereas the kidney weight and kidney hypertrophy were significantly ameliorated compared to the nephropathic rats after PCNPs (80 mg/kg) administration. 

### 3.3. Blood Glucose and Insulin Level

In comparison with the control rats, nephropathic rats experienced a considerable rise in glucose levels and lower levels of plasma insulin due to STZ induction. However, following the treatment of PCNPs in nephropathic rats, the levels of fasting glucose were dramatically reduced; additionally, the plasma insulin levels increased significantly (Figure 1 and Figure 2).

### 3.4. Hb and HbA1c Level

The total Hb and HbA1c values of the untreated nephropathic rats are shown in Figure 3 and Figure 4. Decreased Hb and increased HbA1c levels were remarkably observed in nephropathic rats in comparison with the normal group. After PCNP administration, these levels increased toward near normal.

### 3.5. Blood Urea and Serum Creatinine 

The DN rats showed a considerable rise in urea and serum creatinine related to the control rats shown in (Figure 5 and Figure 6). Compared to the rats in the diabetic control group, the DN rats treated with PCNPs had their urea and serum creatinine levels return to normal. The rats administered PCNPs alone showed no significant change in urea and creatinine compared with normal rats.

### 3.6. Urinary NAG, β-Glucuronidase and Microalbumin Level

Figure 7 shows the effects of PCNPs on the NAG, β-glucuronidase and albumin levels. In contrast to the nephropathic control group, the DN rats treated with PCNPs had their NAG, β-glucuronidase and albumin levels return to near normal. The rats administered PCNPs alone showed no significant change in urinary markers compared with normal rats.

### 3.7. Histopathology of Pancreas

Healthy pancreatic anatomy and exocrine acini were observed around the Langerhans islets at 40× magnification (Figure 8). There was an atrophic islet of Langerhans in the STZ-induced nephropathic group of rats. The degraded nuclei and unstained vacuolated cytoplasm of β-cells were both visible. Additionally, a mass of inflammatory cells was perceived. The pancreas exhibited near-normal cells in the PCNP-treated group of rats, in contrast to the degenerative alterations that were seen in the untreated group. In Figure 8C), sinusoids seemed to be in normal condition. As in the control group of rats, the PCNP alone-treated group appeared normal and had a normal β-cell, showing fortification from the harmful effects of STZ.

### 3.8. Histopathology of the Liver

Histological traits, including radially aligned hepatocytes surrounding the central vein, were visible, and sinusoidal spaces were lined by von Kupffer cells in the rat liver tissue of the control group (Figure 9A). Untreated STZ-induced nephropathic rats had an increase in apoptotic hepatocytes, as shown in a slice of their hepatic tissue (Figure 9B). The STZ + PCNP-treated group of rats shows nearly typical hepatic anatomy, with uniformly organized hepatocytes surrounding the major vein. The sinusoidal spaces resemble nearly those of normal rats (Figure 9C). Hepatocytes distributed laterally around the major vein were seen in the PCNP alone-treated rats. Hepatocytes seemed healthy and active (Figure 9D).

### 3.9. Histopathology of the Kidney

Figure 10A shows the renal histology of the normal group, which revealed that the cortex and medulla were healthy. The STZ-induced nephropathic rats show lining cells of renal tubules, atrophied glomerular capillaries and unstained degraded cytoplasmic areas (Figure 10B). Rats given PCNPs showed normal renal corpuscles and glomeruli predominating in their kidney tissue (Figure 10C). Rats given PCNPs alone exhibited significant renal tissue protection against diabetic alterations (Figure 10D).

### 3.10. Immunohistochemical Expression of TNF-α and IL-6

The immunohistochemistry staining of kidney tissues of the rats in the normal and experimental groups is shown in (Figure 11 and Figure 12). TNF-α and IL-6 are positively expressed in the renal tissues given STZ. Conversely, nephropathic rats treated with PCNPs exhibited a substantial reduction of the TNF-α and IL-6 in comparison with nephropathic rats. The PCNP alone-treated rats presented normal expressions of TNF-α and IL-6.

### 3.11. mRNA Expression of KIM-1 and GLUT-2

The expression of KIM-1 was shown to be considerably higher, and GLUT-2 was shown to be lower, in the kidneys of STZ-induced animals in comparison with the normal group (Figure 13A,B). Conversely, in STZ + PCNP-treated rats, they were prominently reduced compared to the DN rats. In the PCNP alone-treated group of rats, there was no significant disparity in KIM-1 and GLUT-2 expression compared to the normal group.

## 4. Discussion

Diabetes has a high correlation with neuropathy, nephropathy, retinopathy, and other micro- and macrovascular illnesses. Diabetic nephropathy affects between 15–25% of people with type 1 diabetes and between 30 and 40% of people with type 2 diabetes [25]. The kidney is crucial in the body’s process of removing waste materials to maintain the proper balance of vital minerals, fluids, and electrolytes. Blood vessels and kidney cells are harmed by the hyperglycemic state [26]. Pancreatic beta cells are destroyed by STZ due to reactive oxygen species (ROS). The inflammatory and necrotic indicators, including IL-6, TNF-α, and KIM-1, are activated as a result of elevated ROS levels, which raise proinflammatory cytokine levels. This process is frequently focused on fibrosis and inflammation, which leads to diabetic nephropathy [27].

Although pharmacological treatments that retard the growth of diabetic nephropathy are available, there is increased interest in using herbal and nanoparticle-based remedies to treat the advancement of DN and its associated consequences [28]. The antioxidant, anti-inflammatory, thrombotic, antibacterial, and antidiabetic potential of PCNPs has attracted a lot of attention and may serve as a starting point for additional research on diabetic problems such as nephropathy [29,30]. In this study, we evaluated the defensive effects of PCNPs on nephropathy in rats. DN is linked to a considerable loss of body weight, which may be caused by hypoinsulinemia and hyperglycemia [31]. After receiving the PCNPs, the body weight of the DN group of rats considerably increased, demonstrating that PCNPs had a preventative impact on tissue damage brought on by hyperglycemia [32].

Renal hypertrophy may also result from a reduction in the breakdown of renal extracellular components and an increase in protein synthesis [33]. After treatment with PCNPs, the kidney weight and hypertrophy were reduced, which represents the reverse of renal hypertrophy in the nephropathic group. The findings of this study agree with those of earlier studies [34]. 

High blood glucose levels are caused by streptozotocin’s selective induction of necrosis in the beta cells of the pancreas, which results in a loss of insulin production [35]. Thus, in this study, the induction of STZ lead to an increase in blood glucose and decreased insulin levels significantly when evaluated in the normal group of rats. In the co-treatment with PCNPs, the blood glucose level was lowered, and a significant rise in insulin level was evaluated in the nephropathic group, which shows the antihyperglycemic activity of PCNPs. There was no significant change in PCNP alone-treated group compared to the normal group. This may serve as evidence that the findings of this study are consistent with those of earlier research [36].

It is not particularly surprising that the diabetic rats in the current study had lower levels of total haemoglobin because anaemia has been widely reported in diabetics [37]. The nephropathic group of rats exhibited pointedly amplified levels of HbA1c, indicating poor glycemic control. There have been reports of increased glycosylation of several proteins, including haemoglobin, in chronic hyperglycemia [32,37,38]. Total hemoglobin and HbA1c levels in nephropathic rats were brought back to normal after PCNP administration. This might be the result of better glycemic control following the use of the active fraction of PCNPs.

The bulk of urea is eliminated through the kidneys and is a metabolic by-product of protein breakdown [39]. Muscle creatine is frequently tested as an indicator of glomerular function [40], which is an endogenous reservoir of creatinine. As in the earlier study [41], high levels of serum creatinine and urea point to harmful alterations in the kidney. However, the administration of PCNPs significantly reduced these values to normal levels.

The main predictor of early kidney damage is thought to be urine albumin and NAG, the most active lysosomal glycosidase. Its level rises in conditions such as hypertension, nephritis, urinary tract infection, and so on. Urinary NAG has been accompanied by problems in type 2 diabetes mellitus, according to several investigations [42,43]. The lysosomes of tubular cells and the urinary tract epithelial cells produce urinary β-glucuronidase. Microalbuminuria is the most significant clinical sign for the early clinical diagnosis of DN [44,45].

In this investigation, urine NAG, β-glucuronidase and albumin levels, which serve as sensitive indicators of renal tubular damage, were examined to assess kidney tubular functioning [46]. In this present study, higher levels of NAG, β-glucuronidase and albumin activity were seen in DN animals compared to the normal group. This is mostly because lysosomes implicated in oxidative stress-mediated nephrotoxicity were ruptured. Co-treatment with PCNPs lowered the activities of these renal indicators. Thus, PCNPs demonstrated their therapeutic usefulness by defending against or reducing oxidative stress. This may be because of their potential for acting as an anti-inflammatory, as well as their antidiabetic and antioxidant properties. The alterations in urine markers seen in this investigation were similar to those previously described [47].

Comparing the tissue sections of the STZ-induced nephropathic group of rats to the normal group of rats in this study revealed significant disorganized pathological alterations. The islets of Langerhans in the nephropathic group treated with PCNPs appeared to have more or less normal pathology. Additionally, this outcome is consistent with the previous studies [48,49]. In contrast to the normal group of rats, the group given PCNPs alone had a typically observed population of β-cells. These results imply that PCNPs enhance STZ’s ability to shield the pancreas from oxidative stress and injury. 

The STZ-induced nephropathic group of rats treated with PCNPs reimbursed the normal hepatocytes, demonstrating a protective function against hepatic injury. These histological alterations are similar to previous studies [50]. These results imply that PCNPs have a preventative or prophylactic function since the normal structure of the rat liver was observed in the group that received PCNPs alone as treatment.

The renal tissue of the nephropathic group in the current investigation displayed renal corpuscle deformation together with glomerular capillary atrophy. The research of Maric-Bilkan et al. [51], which revealed significant kidney damage and glomerular sclerosis in diabetic rats [52], provides strong support for these findings. Additionally, the PCNP-treated group of rats showed resistance to nephropathic alterations, and the renal corpuscles, glomeruli, and tubules appeared to be in good condition compared to the normal group. According to these results, PCNPs can be used as a preventative approach to reduce the severity of emerging diabetic nephropathy. Based on histopathology, the oral dosing of PCNPs at 80 mg/kg may be a helpful augmentation medication to achieve and/or maintain glucose tolerance and possibly delay the onset of diabetic complications.

Here, we reveal for the first time that infiltration in diabetic glomeruli is ameliorated by the administration of PCNPs. The NF-κB pathway may also be directly activated by the inflammatory markers, creating a positive autoregulatory loop that can lengthen the duration of inflammation and enhance the inflammatory response [53]. The most versatile cytokines for controlling immune modulation, aberrant cell proliferation, and inflammation are TNF-α and IL-6 [38]. In this analysis, the nephropathic group exhibited significantly higher levels of TNF-α and IL-6 expression. However, PCNP therapy dramatically reduced the expression of TNF-α and IL-6. This study demonstrates the anti-inflammatory effects of PCNPs and is consistent with earlier renal illness [54,55].

KIM-1, GLUT-2, and osteopontin mRNA expression have been demonstrated in several renal injury models [55,56,57,58]. Our findings showed that STZ-induced rats had higher KIM-1 expression, which may be primarily due to tubulointerstitial injury. However, the PCNPs significantly decreased the expression of KIM-1 in the kidney, which may be related to PCNPs’ protective impact on the renal tubular cells. Investigations have indicated that STZ-induced nephrotic rats exhibit lower GLUT-2 mRNA expression. After treatment with PCNPs, the expression significantly increased. GLUT-2 mRNA has also been considered as a possible marker of the pancreas.

## 5. Conclusions

This research reveals that PCNPs possess renoprotective activities in diabetic rats with nephropathy caused by STZ and also that PCNPs reduce hyperglycemia-induced oxidative stress, provocative markers such as TNF-α, IL-6, and KIM-1 in renal tissues, as well as GLUT-2 mRNA expression in the pancreas. To a greater extent, research is advised to elucidate the specific underlying mechanisms of PCNPs’ renoprotective action since their usage conserved renal function by reducing the levels of urine indicators such as NAG, β-glucuronidase and albumin levels and reverted kidney damage in diabetic rats. The above findings suggest that co-administration of PCNPs reduces several signs of nephrotoxicity.

## Figures and Tables

**Figure 1 metabolites-12-01166-f001:**
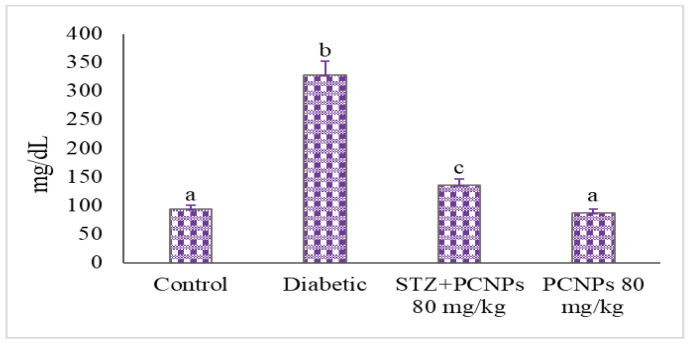
Activities of PCNPs on plasma glucose of experimental and control group. Values are means ± SD for six rats at *p* < 0.05 (DMRT). ^a^ PCNPs alone treated group compared with control group. ^b^ Diabetic group compared with control group. ^c^ Treatment group compared with diabetic control.

**Figure 2 metabolites-12-01166-f002:**
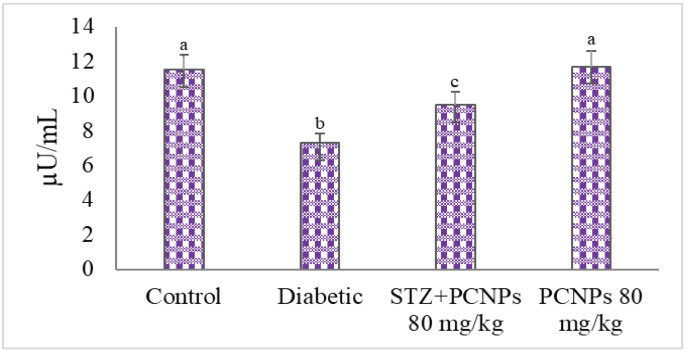
Activities of PCNPs on insulin of experimental and control group. Values are means ± SD for six rats at *p* < 0.05 (DMRT). ^a^ PCNPs alone treated group compared with control group. ^b^ Diabetic group compared with control group. ^c^ Treatment group compared with diabetic control.

**Figure 3 metabolites-12-01166-f003:**
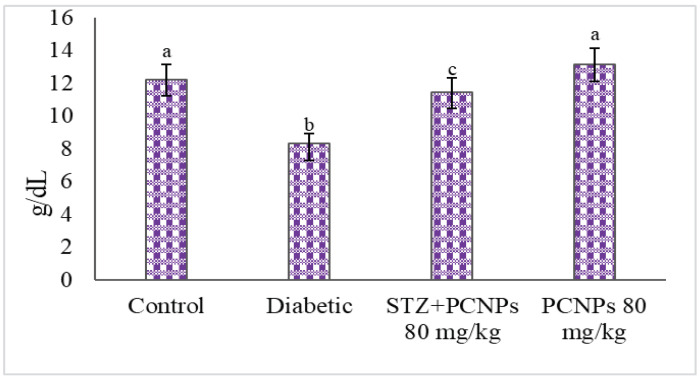
Activities of PCNPs on Hb of experimental and control group. Values are means ± SD for six rats at *p* < 0.05 (DMRT). ^a^ PCNPs alone treated group compared with control group. ^b^ Diabetic group compared with control group. ^c^ Treatment group compared with diabetic control.

**Figure 4 metabolites-12-01166-f004:**
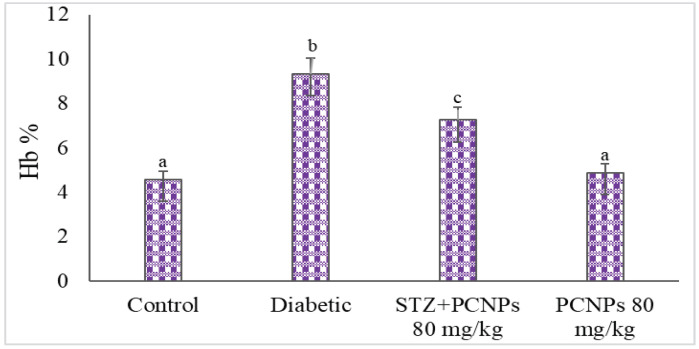
Activities of PCNPs on HbA1c of experimental and control group. Values are means ± SD for six rats at *p* < 0.05 (DMRT). ^a^ PCNPs alone treated group compared with control group. ^b^ Diabetic group compared with control group. ^c^ Treatment group compared with diabetic control.

**Figure 5 metabolites-12-01166-f005:**
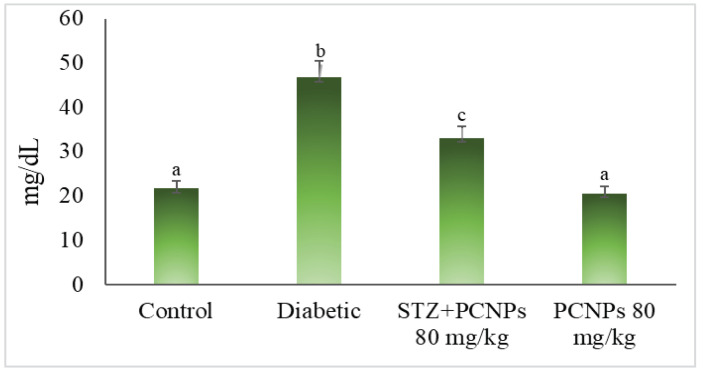
Activities of PCNPs on urea of experimental and control group. Values are mean ± SD for six rats at *p* < 0.05 (DMRT). ^a^ PCNPs alone treated group compared with control group. ^b^ Diabetic group compared with control group. ^c^ Treatment group compared with diabetic control.

**Figure 6 metabolites-12-01166-f006:**
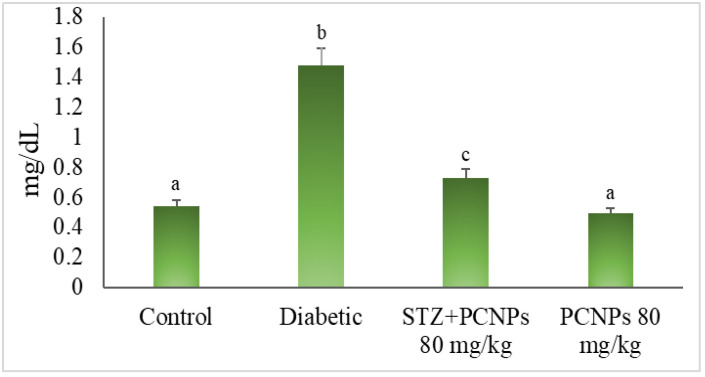
Activities of PCNPs on creatinine of control and experimental group. Values are mean ± SD for six rats at *p* < 0.05 (DMRT). ^a^ PCNPs alone treated group compared with control group. ^b^ Diabetic group compared with control group. ^c^ Treatment group compared with diabetic control.

**Figure 7 metabolites-12-01166-f007:**
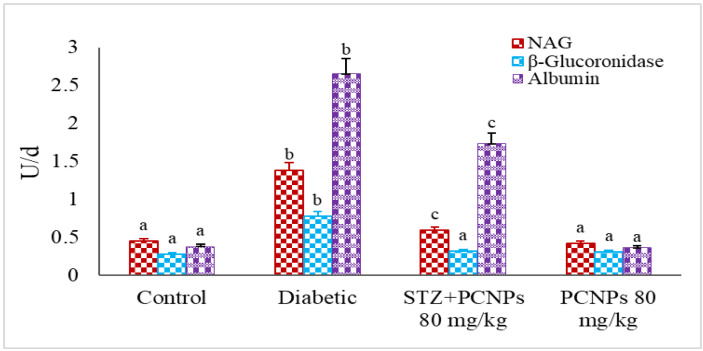
Activities of PCNPs on urinary NAG, β-Glucoronidase and microalbumin of control and experimental group. Values are means ± SD for six rats at *p* < 0.05 (DMRT). ^a^ PCNPs alone treated group compared with control group. ^b^ Diabetic group compared with control group. ^c^ Treatment group compared with diabetic control.

**Figure 8 metabolites-12-01166-f008:**
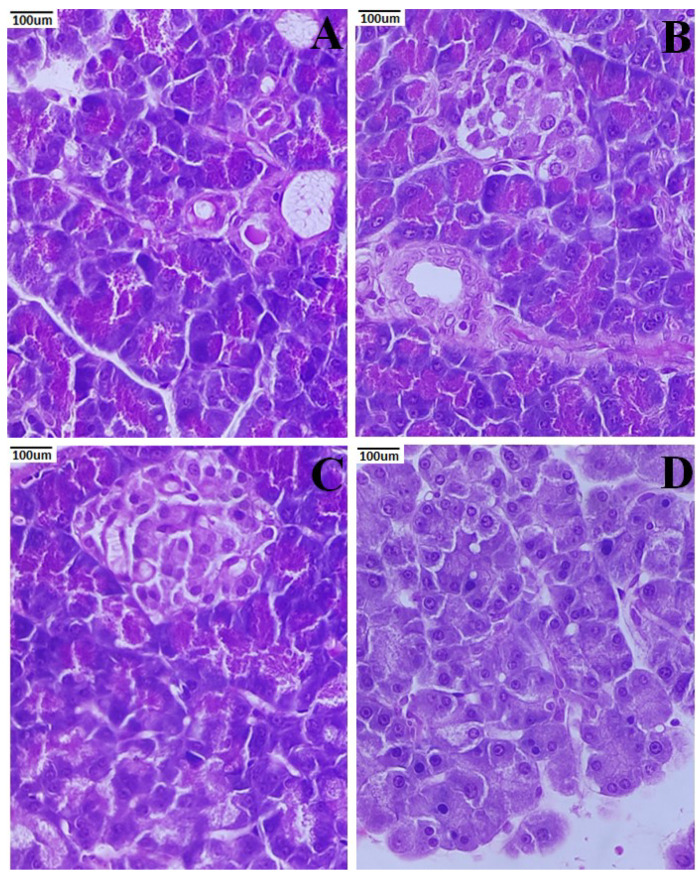
Activities of PCNPs on the pancreatic histopathological picture of DN rats (40×). (**A**) Normal photomicrographic pancreas with islet cells surrounded by acini; (**B**) DN rat showing atrophic pancreatic acini with fatty infiltration; (**C**) STZ + 80 mg/kg of PCNP-treated pancreas shows the regeneration of islet cells with few atrophic acini; (**D**) 80 mg/kg of PCNP only shows the normal architecture of pancreatic cells.

**Figure 9 metabolites-12-01166-f009:**
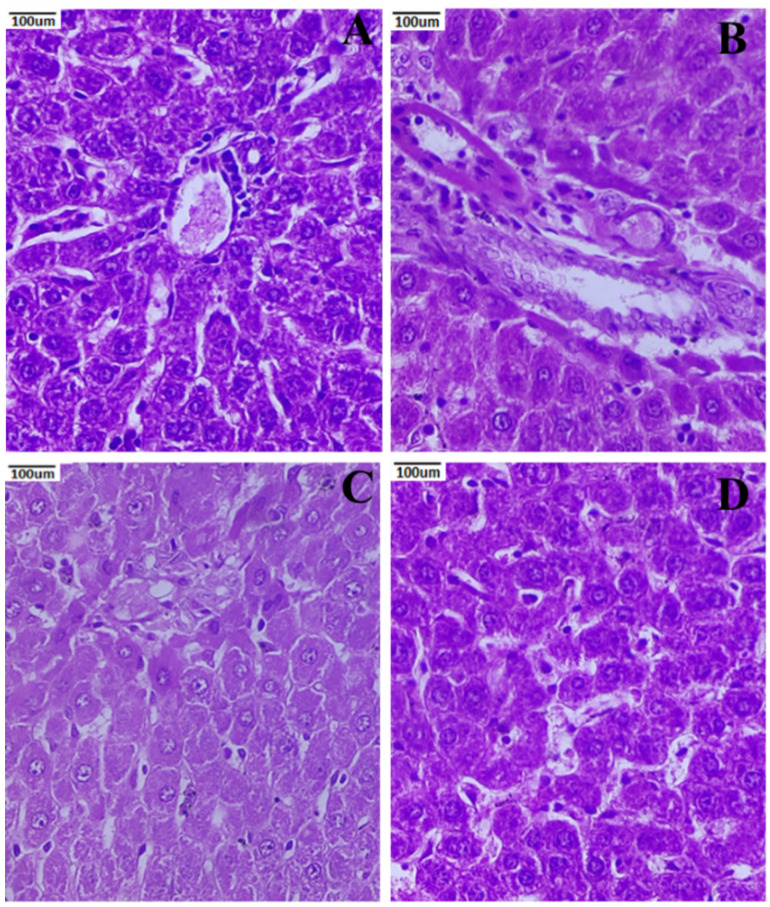
Activities of PCNPs on the liver histopathological picture of DN rats (40×). (**A**) Normal photomicrograph shows hepatocytes arranged in the form of cords and the central vein; (**B**) DN liver showing feathery degeneration, micro- and macrovesicular fatty changes, periportal fibrosis, and vascular congestion; (**C**) STZ + PCNP-treated group of rats shows nearly normal hepatic anatomy; (**D**) 80 mg/kg of PCNPs only shows cells with normal histology.

**Figure 10 metabolites-12-01166-f010:**
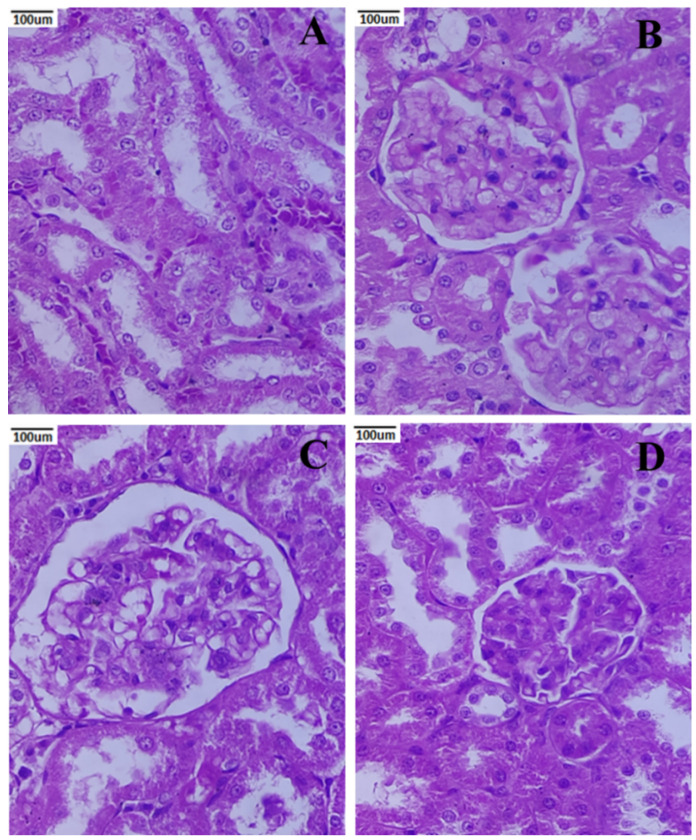
Activities of PCNPs on the kidney histopathological picture of DN rats (40×). (**A**) Normal photomicrograph of kidney shows a regular appearance of the glomerulus; (**B**) DN kidney shows glomerulosclerosis and vacuolation in tubular epithelial cells; (**C**) STZ + 80 mg/kg of PCNP-treated kidney shows the near-normal appearance of glomeruli and tubules; (**D**) 80 mg/kg of PCNPs only shows the normal histology cells.

**Figure 11 metabolites-12-01166-f011:**
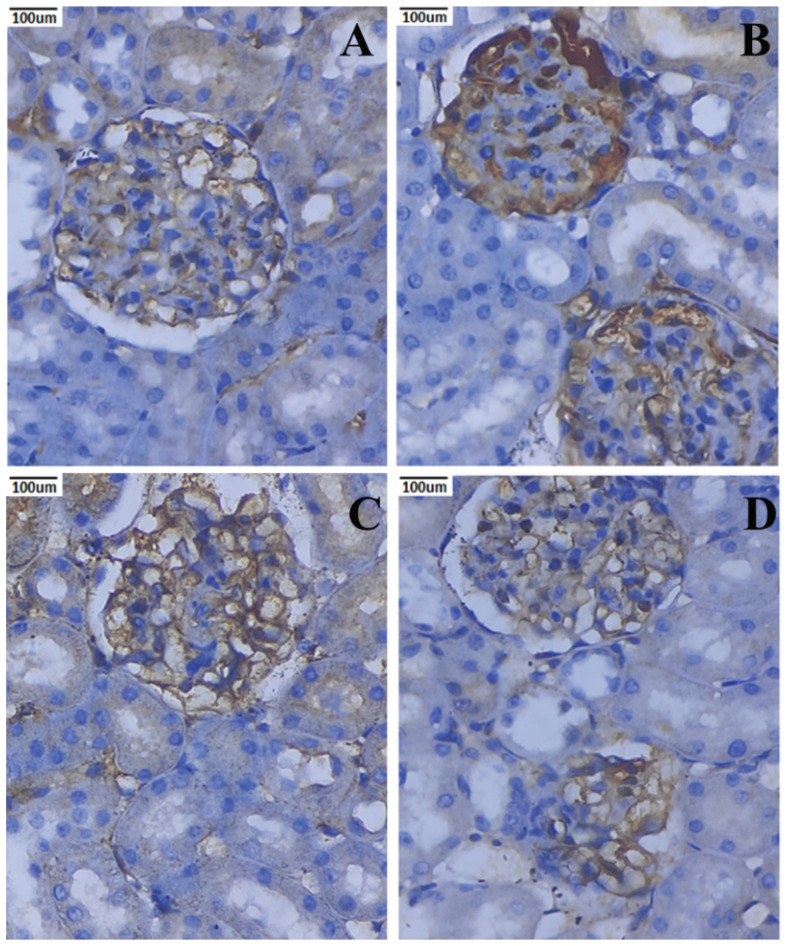
Activities of PCNPs on kidney TNF-α immunohistochemistry of DN rats (40×). (**A**) Control kidney shows a normal expression of glomerulus; (**B**) DN kidney indicates increased expression of TNF-α, showing that there is an increased mesangial cellularity red arrow; (**C**) STZ + 80 mg/kg of PCNP-treated kidney shows the near-normal appearance of mesangial cells and tubules; (**D**) 80 mg/kg of PCNPs only shows the normal expression of TNF-α in mesangial cells.

**Figure 12 metabolites-12-01166-f012:**
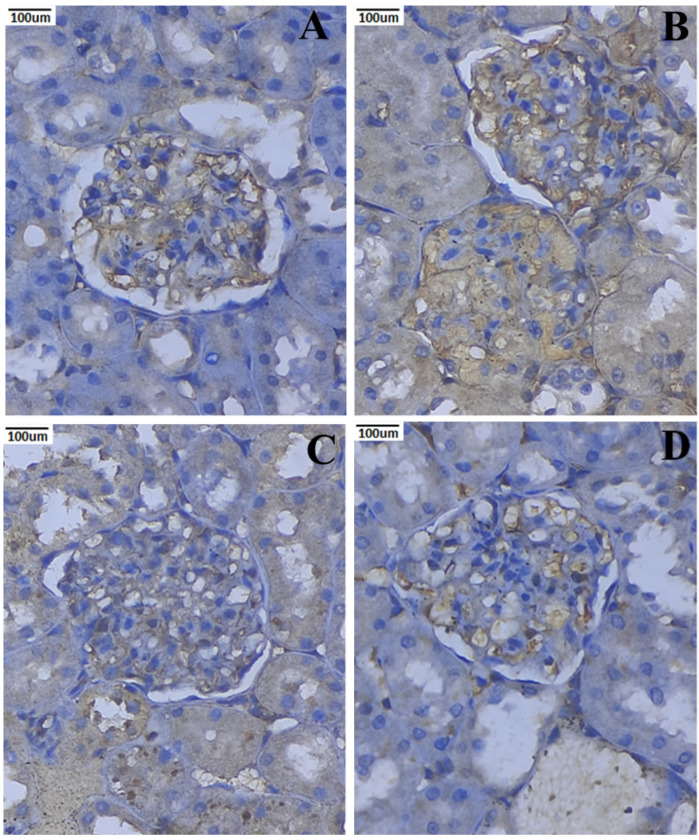
Activities of PCNPs on kidney IL-6 immunohistochemistry of DN rats (40×). (**A**) Control kidney shows a normal expression of glomerulus; (**B**) DN kidney indicates increased expression of IL-6, showing that there is an increased mesangial cellularity at the red arrow; (**C**) STZ + 80 mg/kg of PCNP-treated kidney shows the near-normal appearance of mesangial cells and tubules; (**D**) 80 mg/kg of PCNPs only shows the normal expression of IL-6 in mesangial cells.

**Figure 13 metabolites-12-01166-f013:**
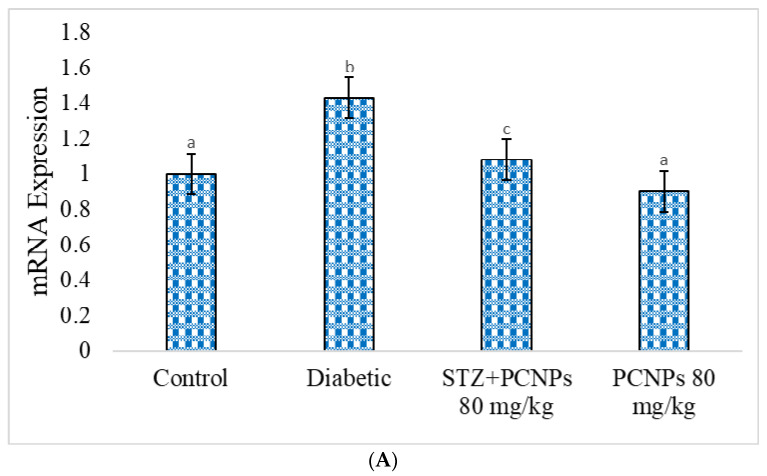
(**A**) Expression of KIM-1 mRNA in renal tissues of experimental and normal rats. PCNPs inhibit the expression of KIM-1 in DN rats. Values are means ± SD for six rats at *p* < 0.05 (DMRT). ^a^ PCNPs alone treated group compared with control group. ^b^ Diabetic group compared with control group. ^c^ Treatment group compared with diabetic control. (**B**) Expression of GLUT-2 mRNA in pancreatic tissues of experimental and normal rats. PCNPs inhibit the expression of GLUT-2 in DN rats. Values are means ± SD for six rats at *p* < 0.05 (DMRT). ^a^ PCNPs alone treated group compared with control group. ^b^ Diabetic group compared with control group. ^c^ Treatment group compared with diabetic control.

**Table 1 metabolites-12-01166-t001:** Oligonucleotide primers for mRNA expression.

S.No.	Gene	Sequence of Primers	Accession Numbers
1	KIM-1	Forward: 5′-GAGTTCATTAGAGCCATTTCCACTCC-3′ Reverse: 5′-GAAAGCCTGTGTCCTGCTCTCTCT-3′	NM_173149.2
2	GLUT-2	Forward: 5′-CTC GGG CCT TAC GTG TTC TTC CTT-3′ Reverse: 5′-TGG TTC CCT TCT GGT CTG TTC CTG-3′	XM_039101783.1
3	β-actin	Forward: 5′-TACAGCTTCACCACCACAGC-3′ Reverse: 5′-TCTCCAGGGAGGAAGAGGAT-3′	NM_031144.3

**Table 2 metabolites-12-01166-t002:** Activities of PCNPs on body weight, kidney weight and kidney hypertrophy index of STZ-induced diabetic nephropathy in rats.

Groups	Body Weight (g)		Kidney Weight (g)	Kidney Hypertrophy Index (%)
	Initial	Final		
Control	225.30	233.50 ^a^	1.85 ^a^	0.79 ^a^
Diabetic	230.50	162.43 ^b^	2.38 ^b^	1.46 ^b^
STZ + PCNPs 80 mg/kg	235.65	193.67 ^c^	2.13 ^c^	1.09 ^c^
PCNPs 80 mg/kg	220.55	230.15 ^a^	1.90 ^a^	0.82 ^a^

Values are given as mean ± SD from six rats in each group. Values not sharing a common superscript letter differ significantly (*p* < 0.05).

## Data Availability

All data are available in the manuscript.

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
