# Peer review of "p-Coumaric Acid Nanoparticles Ameliorate Diabetic Nephropathy via Regulating mRNA Expression of KIM-1 and GLUT-2 in Streptozotocin-Induced Diabetic Rats"

_metabolites, 2022, doi:10.3390/metabo12121166_

Round 1

Reviewer 1 Report

In this manuscript, diabetic nephropathy rats were prepared by intraperitoneal injection of streptozotocin, and after treatment with PCNPS, the results showed that PCNPS could significantly increase body weight, reduce kidney weight, improve blood glucose levels, and improve serum and urine biochemical levels in rats with diabetic nephropathy, but there were some issues, as follows:

1.     What is the age of the rats in weeks and should sexually mature rats be used for the experiment

2.     Streptozotocin injections are generally used to prepare the type 1 diabetic rat model and, of course, the condition of persistent hyperglycaemia can damage blood vessels and kidneys, can just four weeks of diabetes cause nephropathy in rats?

3.     Diabetic nephropathy is associated with elevated urinary microalbumin levels, a key indicator for the diagnosis of nephropathy, which does not appear to be reflected in the experiment

4.     Streptozotocin injection damages the pancreatic tissue of rats, causing atrophy of the islets, but there is no description of the pathological changes in the islets in the pathology results, and the magnification of the four images seems to be different

5.     The article attributes the mechanism of PCNPS treatment in rats with diabetic nephropathy to the regulation of inflammation, but does not have a clear pathway or mechanism of treatment

6.     The description of the pathological findings in the manuscript is not adequate

7.     The discussion is too lengthy and a rewrite is recommended

Author Response

Reviewer 1

Comments and Suggestions for Authors

In this manuscript, diabetic nephropathy rats were prepared by intraperitoneal injection of streptozotocin, and after treatment with PCNPS, the results showed that PCNPS could significantly increase body weight, reduce kidney weight, improve blood glucose levels, and improve serum and urine biochemical levels in rats with diabetic nephropathy, but there were some issues, as follows:

  1. What is the age of the rats in weeks and should sexually mature rats be used for the experiment

Reply: We have received four-week older adult male albino Wister rats then, for acclimation which was maintained another one week, after that we have started the experiment. In this experiment we used sexually matured rats.

  1. Streptozotocin injections are generally used to prepare the type 1 diabetic rat model and, of course, the condition of persistent hyperglycaemia can damage blood vessels and kidneys, can just four weeks of diabetes cause nephropathy in rats?

Reply: Actually, we choose a high dose of streptozotocin (45 mg/kg) based on previous studies; it can cause renal damage with in this experimental period.

  1. Diabetic nephropathy is associated with elevated urinary microalbumin levels, a key indicator for the diagnosis of nephropathy, which does not appear to be reflected in the experiment

Reply: As per the reviewer suggestion parameter Included in this manuscript.

  1. Streptozotocin injection damages the pancreatic tissue of rats, causing atrophy of the islets, but there is no description of the pathological changes in the islets in the pathology results, and the magnification of the four images seems to be different

Reply: As per the reviewer suggestion, the pathology and magnification of the image was corrected.

  1. The article attributes the mechanism of PCNPS treatment in rats with diabetic nephropathy to the regulation of inflammation, but does not have a clear pathway or mechanism of treatment

Reply: Corrected

  1. The description of the pathological findings in the manuscript is not adequate

Reply: Added

  1. The discussion is too lengthy and a rewrite is recommended

Ans: Corrected

Reviewer 2 Report

The present study investigated the protective effects of p-Coumaric Acid Nanoparticles (PCNA) on streptozotocin-induced Diabetic Nephropathy (DN).      Administration of PCNA improved the conventional pathological indicators in DN rats, including inflammatory factors of TNF-α and IL-6 as well as mRNA expressions of the kidney injury molecules KIM-1 and GLUT-2. However, there are tremendous errors in the manuscript, especially the results, which should be revised.

1.      No markers in tables and figures that should be present to indicate differences between groups. Also no “(p<0.05)” in the results description.

2.      Plotting scale in HE staining and immunohistochemical staining figures was questionable, usually 50μm and 100μm are utilized to view the histomorphology. It’s hard to observe the microstructures at scales of 340μm and 800μm.

3.      Apparently, based on your results of the amplification curve, the primers you used here were not correct, as curves did not reach the platform stage after the 40 cycles. Accession numbers missing.

4.      Small errors and mistakes in grammar were everywhere in the manuscript:

Line 21: anti-diabetic, anti-inflammatory, and antioxidant properties of PCNPs, DN development may be considerably down.

Line 30: increased body weight and decreased kidney weight.

Line 31: “The levels of urinary and serum parameters were both lowered by PCNPs” the description was ambiguous.

Line 45: “ESRD” first referred, full name should be present.

Line 134: plasma and serum samples were separated.

Line 145: using the alkaline picrate method [23]. (Qualigens, Mumbai, India).

Line 174: given as mean ±SD.

Line 177: At P 0.05,

Make a thoroughgoing revision of your manuscript and standardize your figures and tables required by the journal.

Author Response

Reviewer 2

Comments and Suggestions for Authors

The present study investigated the protective effects of p-Coumaric Acid Nanoparticles (PCNA) on streptozotocin-induced Diabetic Nephropathy (DN). Administration of PCNA improved the conventional pathological indicators in DN rats, including inflammatory factors of TNF-α and IL-6 as well as mRNA expressions of the kidney injury molecules KIM-1 and GLUT-2. However, there are tremendous errors in the manuscript, especially the results, which should be revised.

  1. No markers in tables and figures that should be present to indicate differences between groups. Also, no “(p<0.05)” in the results description.

Reply: As per the reviewer suggestion, the difference between the groups has been included.

  1. Plotting scale in HE staining and immunohistochemical staining figures was questionable, usually 50μm and 100μm are utilized to view the histomorphology. It’s hard to observe the microstructures at scales of 340μm and 800μm.

Reply: We choose 40x magnification which is 100μm. Typologically, we have mentioned as 340μm. Corrected the same.

  1. 3.      Apparently, based on your results of the amplification curve, the primers you used here were not correct, as curves did not reach the platform stage after the 40 cycles. Accession numbers missing.

Reply: As per the reviewer suggestion, the Accession number of the primers have been added, and based on the previous study we have choose the primers.

  1. Small errors and mistakes in grammar were everywhere in the manuscript:

Reply: As per the reviewer comment, the following suggestion were corrected.

Line 21: anti-diabetic, anti-inflammatory, and antioxidant properties of PCNPs, DN development may be considerably down.

Line 30: increased body weight and decreased kidney weight.

Line 31: “The levels of urinary and serum parameters were both lowered by PCNPs” the description was ambiguous.

Line 45: “ESRD” first referred, full name should be present.

Line 134: plasma and serum samples were separated.

Line 145: using the alkaline picrate method [23]. (Qualigens, Mumbai, India).

Line 174: given as mean ±SD.

Line 177: At P 0.05,

Reviewer 3 Report

The topic that the manuscript addresses is of interest, in addition to showing novelty with the use of nanoparticles, however it is necessary to add relevant information that allows a better visualization of the results obtained. Introduction, it is necessary to indicate information regarding the effects of chitosan on diabetes, as well as the importance of using nanoparticles. Methodology. Indicate how the nanoparticles were obtained, as well as their characterization, size, charge, etc. Indicate the concentration of glucose in the animals to consider them diabetic and at what time this concentration was determined. Results. Unification of graphics, they have different sizes, fonts, etc. unify in the best way In section 3.6 it is necessary to describe the results of beta-glucuronidase level. Conclusions: Based on the results obtained, it is considered that the conclusions are very general, it is possible to be more specific.

Author Response

Reviewer 3

Comments and Suggestions for Authors

The topic that the manuscript addresses is of interest, in addition to showing novelty with the use of nanoparticles, however it is necessary to add relevant information that allows a better visualization of the results obtained. Introduction, it is necessary to indicate information regarding the effects of chitosan on diabetes, as well as the importance of using nanoparticles. Methodology. Indicate how the nanoparticles were obtained, as well as their characterization, size, charge, etc. Indicate the concentration of glucose in the animals to consider them diabetic and at what time this concentration was determined. Results. Unification of graphics, they have different sizes, fonts, etc. unify in the best way In section 3.6 it is necessary to describe the results of beta-glucuronidase level. Conclusions: Based on the results obtained, it is considered that the conclusions are very general, it is possible to be more specific.

Reply: Thank so much for your valuable comments, as per the above suggestions the manuscript has been corrected. Regarding the synthesis and characterization of nanoparticles work was accepted for publication in the Current Pharmaceutical Biotechnology.

Round 2

Reviewer 1 Report

The author has examined and revised the manuscript

Reviewer 2 Report

This review accepts the revised version of this manuscript. 

Reviewer 3 Report

The authors have substantially improved the manuscript and addressed the comments made by the reviewers.